# Measuring Community Resilience and Its Determinants: Relocated Vulnerable Community in Western China

**DOI:** 10.3390/ijerph20010694

**Published:** 2022-12-30

**Authors:** Wei Liu, Jingxuan Zhang, Long Qian

**Affiliations:** 1School of Public Administration, Xi’an University of Architecture and Technology, Xi’an 710055, China; 2Institute of Food and Strategic Reserves, Nanjing University of Finance and Economics, Nanjing 210003, China

**Keywords:** poverty alleviation resettlement, community resilience, relocated households, entropy method

## Abstract

With the full implementation of poverty alleviation resettlement (PAR), the restoration and improvement of the comprehensive living standards of relocated households have received increasing attention from policy researchers. The measurement of resilience and its determinants provides new ideas for PAR at the community level. This article proposes a method for examining community resilience in the context of PAR through a survey of 459 relocated households in western China and uses regression analysis to identify the determinants of community resilience. The results showed that the four dimensions of community resilience, in descending order, included: environmental resilience, economic resilience, management resilience, and social resilience. Income level and livelihood diversification were positively correlated with the community resilience index. Relocation time, relocation type, and resettlement mode were all essential determinants of the community resilience of relocated households. Finally, some suggestions were put forward, such as the need to build an interpersonal relationship network, guide pure farmers and non-farmers to transform into diversified livelihood households, and formulate a unified community action plan and interest protection mechanism so as to provide a reference for decision-making among managers to make decisions.

## 1. Introduction

In the 21st century, due to major climate change, natural disasters, and environmental degradation on a global scale, the question of how best to cope with environmental change has attracted continuous global attention [1]. The resilience and vulnerability of natural and social systems in special regions have become a central issue for the science of sustainable development [2]. People living in poverty-stricken areas in developing countries have to face multiple challenges of ecological improvement, poverty eradication, and social development [3]. Although the relocation and resettlement of households to ecologically better settlements should be a last resort, this approach offers a glimmer of hope for struggling farming households [4]. To improve ecosystem services and mitigate natural disasters caused by ecosystem degradation in an effort to enhance human well-being, China’s Shaanxi Province launched a relocation and resettlement program in 2011 [5]. There is a greater spatial distance between the relocated communities and their original homes in inhospitable areas of Shaanxi province, and the material environment of farmers has changed dramatically after relocation [6]. Despite improvements in housing and infrastructure, living and productive resources have been reallocated, incomes have generally declined, and farmers’ former productive activities are no longer relevant [7]. In the period after relocation, farmers’ living standards not only failed to reach the policy’s goal but also showed a trend of relative decline [8]. Therefore, there are certain issues that policymakers and scholars need to discuss together, such as how to shorten the time it takes for relocated households to return to their original living standards, how to accelerate the development of relocated household communities towards common prosperity, and how to achieve the goals of the poverty alleviation resettlement (PAR) project. It is increasingly important to study how to improve the resilience of poverty alleviation relocated household communities in inhospitable areas.

The poverty alleviation and relocation project started in 2011 and ended in 2020, lasting for ten years in China. In the early stages of implementing the relocation policy to alleviate poverty, research scholars devoted themselves to dismantling the incompatible development model that had been developed to address regional ecological resources and the concentrated population [9,10,11]. The focus of policy implementation in the mid-term of resettlement has changed to supporting the special industries in the resettlement area, maintaining the stable development of farmers’ living standards, and improving the economic level. Research on PAR and the direction of such research shifted from environmental protection toward livelihood strategies [12], different resettlement modes [13], poverty reduction measures [14], and livelihood vulnerability reduction [15]. Nowadays, China’s PAR project has entered the post-relocation era, and the focus of PAR research falls on improving the living standards and sustainable development of relocated households [16], reinventing social networks [17], upgrading public service facilities in resettlement sites and fully protecting the legal rights and interests of relocated people [18,19]. The construction of relocation projects for poverty alleviation and resettlement has been fully completed, and government departments attach great importance to a combination of poverty alleviation policies through relocation and the comprehensive reform of towns and cities as well as new urbanization [20]. Under the background of poverty, the work of PAR focuses more on improving the living conditions of displaced families; however, less attention has been paid to integrating households into their new area, recovering income levels, and improving labor skills. Therefore, relocated households have low levels of stability and are at risk of falling back into poverty. In response, the 19th Communist Party of China National Congress put forward a rural revitalization strategy to solve the unbalanced and inadequate rural development problem. The government will attempt to consolidate the results of poverty alleviation, articulate rural revitalization [21], improve the livelihood resilience of relocated households [22] and enhance the community resilience of resettlement sites. 

Although research has examined the living conditions of resettlement households before and after disaster-related relocation [23,24,25,26,27], studies on community resilience in poverty-stricken areas of China are relatively rare [28]. A community is a large collective connected by social organizations or social groups that gather together; it can be regarded as a homogeneous social structure and an important urban risk control unit [29]. Resilience was originally used to depict the ability of a system or material to resist disturbances and maintain its basic functions in a specific state [30,31]. Resilience, in the social system’s domain, is understood as the ability of an organization, individual, or community to adapt, resist, transform and recover from the effects of persistent stress or disruptive events in an effective and timely manner, and to recover or adjust accordingly [32,33]. Community resilience is a key indicator of social sustainability [34], which refers to how community members adjust their behavior and perceive changes in the social environment based on past experiences and available knowledge in order to achieve beneficial effects that collectively improve community functioning and well-being [35]. Community resilience is a dynamic process [36], and its components may sometimes weaken or strengthen the pathways to resilience, but the basis for community survival and development is that the overall state is balanced and good [37]. Susan proposed that the subcomponents of community resilience are economic resilience, social resilience, institutional resilience, and infrastructural resilience [38]. Identifying community resilience and its determinants can provide a basis for developing adaptive management responses and exploring community management practices [39]. The ability of community residents to adapt to changes in their living environment can be improved by developing diverse livelihood options and strengthening self-organization [40].

As mentioned earlier, research on community resilience is emerging worldwide, and the empirical analysis framework in one particular site may not apply to other areas due to regional characteristics and relocation features [41]. Exploring community resilience and its determinants in resettlement areas is an urgent matter for Ankang Prefecture, Shaanxi Province, China, where the resettlement project has just been completed, and this paper helps to fill this research gap. Therefore, this article measured the community resilience of rural households that have relocated from poverty-stricken areas in Shaanxi Province and quantitatively analyzed the determinants of community resilience. First, community resilience is divided into four dimensions: economic resilience, social resilience, management resilience, and environmental resilience, and the relevant variables that can represent the characteristics of each dimension are identified and weighted to calculate the community resilience index. Next, a comparative analysis of the community resilience of different types of relocated households is conducted. Finally, a multinomial logit regression model is used to empirically analyze the determinants of the community resilience of relocated farm households.

## 2. Materials and Methods

### 2.1. Study Area and Data Source

Most of the areas involved in the poverty alleviation resettlement project are ecologically fragile areas, seismically active zones, geological disaster-prone areas, and areas where the promotion of production or employment through local assistance is still unable to lift the farmers out of poverty. The focus of the project is to help poor households with relatively deep poverty. This study uses Ankang Prefecture, Shaanxi Province, which is located in the southeastern part of Shaanxi Province, as the case study area (Figure 1). It has jurisdiction over one district and nine counties and is an important water source area of the middle route of the South to North Water Transfer Project. The relocation of 268,400 households and 941,000 people from inhospitable areas in Ankang involves complex causes of poverty among farmers in this region, a serious return to poverty, and high livelihood vulnerability [42]. At present, the PAR project has moved into the follow-up stage of support, and the government’s focus has shifted from poverty alleviation to rural revitalization. The rural revitalization strategy is characterized by universality, integrity, and economy, which can more comprehensively activate the endogenous impetus in rural areas, provide more stable and sustainable development opportunities for the poverty groups, and deepen the achievements of poverty alleviation [43].

The research samples were selected from areas with concentrated ecological policies and prominent nature conservation problems, as well as from a pool of relocation projects. Based on abundant field investigation data, the research team took the administrative villages and resettlement communities of Ziyang County, Hanbin District, and Ningshan County of Ankang Prefecture, as well as 10 villages and towns around them, as the research area. The empirical research section of this paper uses data related to a special investigation of farmers’ livelihoods in Ankang, southern Shaanxi [44]. The data used in this study were obtained from a field survey conducted by the subject team in 2015 among farm households in the case area, which focused on household heads and their spouses aged 18–65 years and used household questionnaires and semi-structured interviews. After the questionnaire interview, information input, data correction, and other work, 657 questionnaires were collected, with an effective recovery rate of 98.06%. The questionnaire survey involved two groups of relocated households and non-relocated households. The study of community resilience mainly focused on relocated households, so 198 samples of non-relocated households were removed, and 459 relocated samples were finally obtained as the research objects of this study. The contents of the questionnaire survey included natural capital such as land, basic family situation, family production and consumption behavior, relocation time type, household social and demographic characteristics of households, family livelihood activities and income sources, etc. Overall, the survey sample had good representation.

### 2.2. Method

In order to measure community resilience, it is necessary to analyze the constituent dimensions of community resilience first and then build an index system to measure it. Community resilience studies focus on the balance between community economic productivity, social demands, and environmental health. This study draws on international research results and combines the actual situation of the research region to analyze the community resilience of relocated households from the four dimensions of economic resilience, social resilience, management resilience, and environmental resilience [45] (see Table 1).

Among the four dimensions of the criterion layer of the resilience measurement system, economic resilience refers to the ability to promote economic growth and reduce economic leakage [46], and it encompasses relocation housing subsidies, per capita annual net income, income diversity, and the labor force ratio. For relocated households, housing subsidies can effectively alleviate the economic losses caused by poverty alleviation resettlement. Annual per capita income reflects the savings and purchasing power of relocated households. Relocating households with a higher saving capacity means that these households can enjoy more sustainable livelihoods while relocating households with a higher purchasing capacity contribute to a higher current quality of life [47]. Therefore, the level of annual net income per capita is a measure that can be used to represent sustainable livelihoods and the economic status of families. Income diversity reflects the richness of relocated households’ economic activities [48]. The income sources of households relocated as part of a poverty alleviation scheme in inhospitable areas mainly include agricultural income, working wages, and non-agricultural business turnover. The labor force ratio is the number of people in the labor force in a given household as a proportion of the total household size. An increase in the value of labor results in a lower demographic burden, which creates more household wealth. Households that are more resilient to risk are better able to recover from shocks, thus reducing the likelihood that they will fall into poverty [49]. Therefore, the labor force ratio is a determinant influencing family adaptation and risk resistance. 

Social resilience is related to social trust, support, and belonging, and social capital is essential for social resilience [50,51]. In this study, social resilience was characterized by social help, social relationships, and learning and training opportunities available to community members. Social help refers to the number of family and friends to whom households can turn when they encounter temporary expenditure difficulties with respect to their livelihood activities. The number of families that can be relied upon can be used to measure the scale of social networks. Social relationships refer to the number of relatives and friends who remain in close contact with the household, which can be calculated by the act of exchanging gifts and sending and receiving text messages with friends and relatives. Social learning refers to the average number of years of education in a peasant household. More time and energy are invested in education and training to gain not only rich cultural knowledge and professional skills but also to encourage members of social learning to communicate with each other. In addition, the average number of years of education in relocated households reflects the importance a peasant household attaches to learning. People with higher education are more receptive and adaptable, better at seizing opportunities, and more prepared for risks arising from opportunities [52].

Management resilience, also known as institutional resilience, depends on the capabilities and level of community management and community rights sharing and is represented by policy support and the level of social security and government assistance. Policy support refers to the number of policies issued by the government that can improve farmers’ economic conditions and supports their lives. The level of social security refers to the degree to which the policies designed to protect the rights and interests of the place of residence are implemented. The government assistance level refers to the government non-agricultural management activity reward or allowance.

Environmental resilience includes physical environment resilience and natural ecological environment resilience, which are quantified by accessibility to public facilities and ecological subsidies, respectively [53]. Accessibility of public facilities refers to the accessibility of public services in a new place of residence, which depends on the number and scale of public services, transportation conditions, and population distribution. The layout of equal and adequate public service facilities can improve the quality of life of relocated households and deepen their sense of identity in the destination [54]. Ecological subsidies are a primary goal of environmental protection and restoration and are based on improving the livelihoods of households, an example of which is the subsidy issued by the government for converting farmland to forest and ecological forests in the interest of public welfare.

### 2.3. Entropy Evaluation Method

The essence of entropy is the degree of chaos inherent in a system [55]. The smaller the entropy index, the more information it provides. The more important its role in a comprehensive evaluation, the higher its weight. The advantage of the entropy method mainly lies in the fact that the weight is determined according to the degree of difference, and the weight is determined according to the existing objective information of the data, which can eliminate the interference of human factors, avoid the subjectivity of weight determination, and produce a real and objective evaluation of the system [3,56]. The calculation steps are as follows:

Step 1: Determine the metrics. In the case of n samples and m indicators, it is the value of the jth indicator of the ith sample (i=1, 2, …, n;j=1, 2, …, m).

Step 2: Normalization of indicators.

Positive indicators:(1)$$X_{ij}=\frac{X_{ij}-\min\left\{X_{1j},\;\ldots ,X_{nj}\right\}}{\max\left\{X_{1j},\;\ldots ,X_{nj}\right\}-\min\left\{X_{1j},\;\ldots ,X_{nj}\right\}}$$

Negative indicators:(2)$$X_{ij}=\frac{\max\left\{X_{1j},\;\ldots ,X_{nj}\right\}-X_{ij}}{\max\left\{X_{1j},\;\ldots ,X_{nj}\right\}-\min\left\{X_{1j},\;\ldots ,X_{nj}\right\}}$$

Step 3: Calculate the proportion of the ith sample value of the item to the index.
(3)$$P_{ij}=\frac{X_{ij}}{\sum_{i=1}^{n}X_{ij}},\;i=1,\ldots ,n;j=1,\ldots ,m$$

Step 4: Calculate the entropy value of item j.
(4)$$e_{j}=-k\sum_{i=1}^{n}P_{ij}\ln\left(P_{ij}\right),\;j=1,\ldots ,m$$

Step 5: Calculate information entropy redundancy.
(5)$$d_{j}=1-e_{j},\;j=1,\ldots ,m$$

Step 6: Calculate the weight of each indicator.
(6)$$w_{j}=\frac{d_{j}}{\sum_{j=1}^{m}d_{j}},\;j=1,\ldots ,m$$

Step 7: Calculate the overall score of each sample.
(7)$$s_{j}=\sum_{j=1}^{m}w_{j}P_{ij},\;\;j=1,\ldots ,m$$

The weight of each index $w_{j}$ can be objectively measured by using the entropy method. The closer the value of $s_{j}$ to 1, the more scientific the entropy method is used.

### 2.4. Classification Standard of Community Resilience

The K-means clustering analysis was conducted to divide the community resilience index of relocated households into three classification levels: low, medium, and high, and the significance was tested. Because this classification passed the significance test, it can be regarded as reasonable to a certain extent. The classification results of 459 survey datasets on the resilience of rural households showed that the number of rural households with low, medium, and high levels of community resilience index was 133, 196, and 130, respectively, accounting for 29.0%, 42.7%, and 28.3% respectively. The overall recovery status of relocated households from inarable areas showed a normal distribution (See Table 2). 

## 3. Results

### 3.1. Descriptive Statistics of the Sample’s Basic Profile

Resettlement modes of PAR can be divided into centralized resettlement, scattered resettlement, self-determined resettlement, and other resettlement modes. Descriptive statistics were calculated based on the characteristic values of variables with different resettlement modes in the survey sample, and variance test results were shown in Table 3. It was found that the housing subsidy and the number of households that could seek assistance from centralized resettlement households were the highest, indicating that the government provided more financial help to centralized resettlement households in the process of implementing the poverty alleviation policy in inhospitable areas. Moreover, little change was detected with regard to the neighbors of centralized resettlement households, whose relationship networks appear more stable than before relocation. Among all relocation types, the average number of years in education, income diversity, and the proportion of the labor force composed of scattered resettlement households was the highest. This shows that the age structure and education structure of the labor force of scattered resettlement households are better since they can engage in more diversified economic activities. Compared with centralized resettlement and scattered resettlement, the per capita net income of self-determined resettlement households was the highest, but the number of families that could seek help was the lowest. 

### 3.2. Measuring Results of Community Resilience among Relocated Households

The four dimensions of community resilience were analyzed by referring to the sub-dimension box plot (as shown in Figure 2). The median economic resilience index is located in the middle of the box. The overall distribution is uniform, and the span is the largest, showing that this type of resilience index is scattered toward the poles. The median of the social resilience index is the lowest, and the cut-off points of the two segments are also the lowest compared with other dimensions of the resilience index, indicating that the social resilience of relocated households in the research area of research is poor. This finding suggests that it is difficult to build social networks. The management resilience index has a skewed distribution, and the median is inclined to the lower quartile, showing the characteristics of low concentration distribution. The upper and lower truncation points of the environmental restoring force index are close to the box body and have a skewed distribution, indicating that the level of the environmental restoring force is generally high and concentrated.

### 3.3. Analysis of Community Resilience of Relocated Households

According to the classification method of different relocation characteristics proposed by Liu et al. [22], based on the established community resilience evaluation index system, to analyze the resilience index of relocated household communities and the classified scatter diagram and boxplot were drawn.

As shown in Figure 3, significant differences were observed in community resilience among relocated households in accordance with different relocation types. On the whole, the resilience indexes of different types of communities are mainly concentrated between 0.4 and 0.7, indicating that the communities in the survey area are highly resilient. The community resilience index of poverty alleviation households is relatively concentrated. The internal differentiation of ecological restoration households is obvious, and the overall index is high. The distribution range of the resilience index of project-induced communities comprised of relocation households is relatively concentrated and generally low, indicating that the resilience index of these is relatively close, but the recovery degree is poor. The resilience index corresponding to disaster-related communities comprised of relocation households has the largest distribution span. It is concentrated in the middle, dispersed in the poles, and has more specific values.

The sample size of the centralized resettlement is the largest, and the distribution span of the community resilience index is large, dense in the middle, and scattered at both ends. The distribution of the community resilience index in scattered resettlement communities is unbalanced. The distribution span is the largest. Moreover, the internal differentiation is obvious, indicating that scattered resettlement households are weakly correlated with each other, so the recovery degree is different. The overall resilience index of self-determined resettlement communities is relatively low, with the main body concentrated between 0.3 and 0.7. There are relatively few benefits associated with relocation policies for self-determined resettlement households. Therefore, overall, the community recovery effect of self-determined resettlement households is weaker than that of government-led resettlement (See Figure 4). 

The distribution span of the community resilience index of relocated households with a relocation time of less than 3 years is relatively large. This indicates that the level of resilience of relocated communities with a shorter relocation time is unstable, and there are obvious differences among different relocated households. The community resilience index of relocated households with a relocation time of 3 to 5 years is scattered, but the overall value is high, indicating that the recovery of relocated communities is better when the relocation time is longer. The resilience index of relocated communities that moved more than 5 years is generally low. According to the survey, this phenomenon is due to changes that have taken place in the poverty alleviation policy over the years. The relocation policy 5 years ago was somewhat inadequate and, in large part, failed to provide effective support to encourage community recovery with regard to relocated households. Therefore, the earliest group of relocated households did not have a higher community resilience index than those who had relocated 3–5 years ago (see Figure 5). 

The median community resilience index of high-income relocated households were the highest, and the overall community resilience index has a skewed distribution, indicating that high-income relocated communities are more resilient. The community resilience index of middle-income relocated households generally showed a normal distribution with strong data symmetry. The median resilience index of low-income relocated communities is located in the middle of the box. The distance between the upper and lower cut-off points and the box is equal, and the distribution is relatively uniform. Overall, the community resilience index is positively correlated with income level (see Figure 6).

The median of the resilience index of the relocated community from the village was inclined to the upper part of the box body, and the cut-off point of the upper part was close to the box body, presenting a high concentration distribution. The median of the community resilience index of relocated households from neighboring villages is located in the lower part of the box, and the lower truncation point is close to the box, showing a low concentrated distribution. On the whole, the community resilience index of relocated households from the village is higher than that of neighboring village relocation households, which indicates that the closer the distance of village-level relocation, the higher the community resilience index. The median community resilience index of relocated households from neighboring towns is the highest and skewed to the upper quartile. The community resilience index of relocated households from other places is evenly distributed (see Figure 7).

In terms of the community resilience index of relocated households with different livelihood types, the median of the community resilience index of non-peasant households tends to be the lower quartile, showing a skewed distribution. The median community resilience of pure farmers tends to the lower quartile of the box, with the upper cutoff point farther from the box and the lower cutoff point closer to the box. Compared with the other two livelihood types, the median and upper and lower cut-off points of the community resilience index of diversified livelihood households are the highest, and the median is located in the middle of the box, presenting a high and balanced distribution, indicating that the diversification of livelihood types is conducive to the recovery of farmers’ livelihoods and communities (see Figure 8).

### 3.4. Determinants of Community Resilience of Relocated Households 

Based on the above findings, this study empirically analyzes the determinants influencing the resilience community of relocated households in poverty survival environment areas with the help of a multinomial logit regression model. The resilience community of relocated households was set as the explanatory variable, and the relocation time, relocation type, and resettlement mode were selected as relocation characteristic variables. Household size, income diversity, net income per capita, age of the household head, knowledge of relocation policies, the possibility of loans, and monthly communication costs of household members were used as reference variables.

The regression results in Table 4 show that relocation time had a significant effect on community resilience. The log-likelihood value in the model was −426.315, the chi-square test value was 0.1232, and the significance level was 0.000 (<0.05), indicating that the equation is significant overall. Short-term relocation was used as a reference group, and the data showed that community resilience decreased as the time of relocation increased. The relocation time increased with less change in external natural conditions, while the relocation policy and support mechanism improved. This can indicate that the recently introduced policies are more conducive to the restoration of relocated households’ living standards, and research on relocation mechanisms has achieved remarkable results.

The relocation types were divided into ecological restoration households, poverty alleviation households, project-induced relocation households, disaster-related relocation households, and other types of relocation households. Ecological restoration households were selected as the reference group in the regression process. Table 5 shows that the type of relocation had an important effect on community resilience. With constant reference variables, relocated households triggered by poverty ecology had the best community resilience. This was followed by poverty reduction and disaster-related relocated households, while project-induced relocated farmers had the weakest level of community resilience.

The resettlement modes were categorized into centralized resettlement, scattered resettlement, self-determined resettlement, and another type of resettlement modes. Table 6 shows the multinomial logit regression model with community resilience as the explanatory variable and resettlement mode as the relocation characteristic, where the centralized resettlement mode was selected as the reference group. The significance level was 0.000 (<0.05), and the equation was generally significant. The regression results indicated that the resilience of relocated households with centralized resettlement was the strongest, and the community resilience level of scattered resettlement and self-determined resettlement households was weaker.

## 4. Discussion

Compared with the extant literature, the potential contribution of this study is that it can provide a referential framework for scholars and resettlement communities themselves who wish to measure community resilience, and it also proposes more refined solutions to community development dilemmas. Previous research has focused on the determinants influencing livelihood resilience [57,58] and determinants that significantly affect residents’ livelihood strategies [59,60,61]. Based on the poverty alleviation resettlement project, this study explored the link between community resilience and relocation projects by carrying out an empirical analysis. It further identified the determinants that influenced the community resilience of relocated farm households. In addition, numerous scholars have linked community resilience with disasters [62,63], climate change [64,65], and epidemics [66,67,68] through quantitative analyses. This study complements research modes that focus on the quantification of community resilience and contributes to improving the community living standards of rural households relocated from poverty-stricken areas. The community resilience of relocated households describes the ability of relocated farming communities to withstand and recover from the adverse impacts of drastic changes in the physical environment and the reallocation of livelihood and production resources [69,70]. While reshaping and changing the structure and context of social vulnerability with regard to relocation and resettlement areas, community resilience can guide the livelihood adaptation of households [70]. In the whole research system, the framework construction of the community resilience indicator system is the foundation of research, and resilience research focuses on how to maintain sustainable development in the face of vulnerability in relocated communities and ultimately implements the livelihood adaptation of relocated households as a social group. Its logical progression elucidates the interactions between environmental change, social structures, and actors. It promotes the intersection and integration of the fields of social vulnerability, resilience, and adaptation research [71,72,73,74,75], while enhancing the sustainable development of relocated communities [76,77]. 

Relocation for the purposes of poverty alleviation has been a significant approach to managing environmental change, and it is a method that aims to improve the quality of life of community residents. In terms of resettlement methods, communities with centrally resettled households have the best level of recovery. The descriptive statistics showed that the highest number of assisted households is available to centrally resettled households. This is because, in most rural areas of China, many relatives and friends live in the same community, forming a cluster of shared benefits and shared risks [78,79]. Similar behaviors and traits bring people closer together [80]. The planning and construction of infrastructure in centrally located communities are usually better than in other communities, and relocation housing subsidies are higher. Industrial parks are the usual choice of location for such resettlement sites, as they offer easier access to business information and employment resources, which helps households to broaden their income channels. In addition, the establishment of centralized resettlement communities makes it easier for the government to arrange centralized guidance, job training, and increased opportunities for community residents to develop business activities and work outside the home [81]. The results of previous studies showed that self-determined resettlement had a significant negative effect on the income diversity index, while non-self-determined resettlement had a significant positive impact on the poverty rate [82]. Although scattered resettlement households and self-determined resettlement households have more autonomy, only a portion of economically well-off families have a better recovery level [5], and the overall recovery effect is lower than that of centralized resettlement. 

Our findings suggest that, for the duration of relocation, medium- and long-term relocated communities have lower levels of recovery than short-term relocations. Relocated households had not fully recovered from the experience five or more years after relocation. Similar results have been found in previous studies. On the one hand, because poverty-stricken households relocate to resettlement areas, relevant government departments may perceive that the relocation policy to alleviate poverty is coming to an end, and support is weakened or no longer offered as follow-up support [83]. On the other hand, there are many frail elderly people and left-behind children among the relocated households, and it is difficult for them to find a new way of life that would allow them to adapt to the new environment [3]. Therefore, the continued payment of state subsidies is necessary to accelerate the recovery of relocated communities. There are differences in the impact of the type of relocation on community resilience. Farmers relocated for ecological reasons have a higher community recovery capacity. This Nguyen study is consistent with the fact that, as the goal of eco-settlement gradually shifts from environmental protection to poverty alleviation, many researchers have found that it can greatly reduce poverty by assisting rural households to stay away from vulnerable living environments and earn more economic income from non-farm sources [84]. Households affected by natural disasters have cultivated land areas for livelihoods after relocation, and they are often forced to choose a form of non-farming employment, which leads to more economic income [8]. 

This study addressed a gap in existing research on PAR by focusing on community resilience, but it also had some limitations. First, there are no unified regulations for the selection of resilience indicators at home and abroad, and the subdivision of community resilience dimensions is vague. The 12 variables extracted in this study were composed of survey samples related to the economy, management, society, and the environment, and the selection of variables was somewhat subjective. Second, the survey was restricted to Ankang Prefecture in the south of Shaanxi Province. It did not take into account the data of the central Shaanxi plain and northern Shaanxi regions. In addition, the selected data reflected sectional rather than panel data, so no comparison was carried out between different geographical locations and different development stages. Further research should expand the scope of the study area, obtain relocated household data in different periods, improve the degree of data diversification, and dynamically monitor the recovery of relocated household communities.

## 5. Conclusions

This paper incorporated community resilience into the study of relocated households. To gain greater insight into the resilience of relocated household communities, a measurement index system was developed based on four dimensions: economic resilience, social resilience, management resilience, and environmental resilience. Ankang Prefecture in southern Shaanxi was selected as the case research area. Based on the household questionnaire data, the entropy method was used to assign weights to the components of the resilience of relocated household communities, then the community resilience index was calculated, and the multiple linear regression method was used to explore which determinants had a large impact on the resilience of relocated household communities.

The measurement results of the four dimensions of community resilience included environmental resilience, economic resilience, management resilience, and social resilience, from high to low. This shows that, compared with the accumulation of material and capital, a disrupted interpersonal network is difficult to rebuild, and it takes a long time to recover. Moreover, income level and livelihood diversification were positively correlated with the community resilience index. Relocated households with high-income levels generally have a wider range of livelihood sources, and it takes a shorter period of time for their standard of living to return to their previous state. However, the adaptability and livelihood security status of poverty households with a single livelihood mode after the relocation is worrying. Through the analysis of the determinants influencing the resilience of relocated household communities, it can be seen that relocation time, relocation type, and resettlement method were all identified as determinants that influenced the resilience of relocated household communities. With the same level of policy support, the longer the households are relocated, the better the community recovers. Significant differences were observed in the community resilience of relocated households of different relocation types. The scale benefits generated by centralized resettlement help households obtain additional follow-up support and avail of more support facilities. The community resilience of centralized resettlement households is generally superior to scattered resettlement households and self-determined resettlement households. Therefore, policymakers should take the centralized resettlement model as the policy orientation. The government and community service departments should cooperate with higher administrative departments to develop a unified community action plan, interest protection mechanism, and follow-up support policies to prevent households from feeling isolated and helpless after self-determined resettlement.

## Figures and Tables

**Figure 1 ijerph-20-00694-f001:**
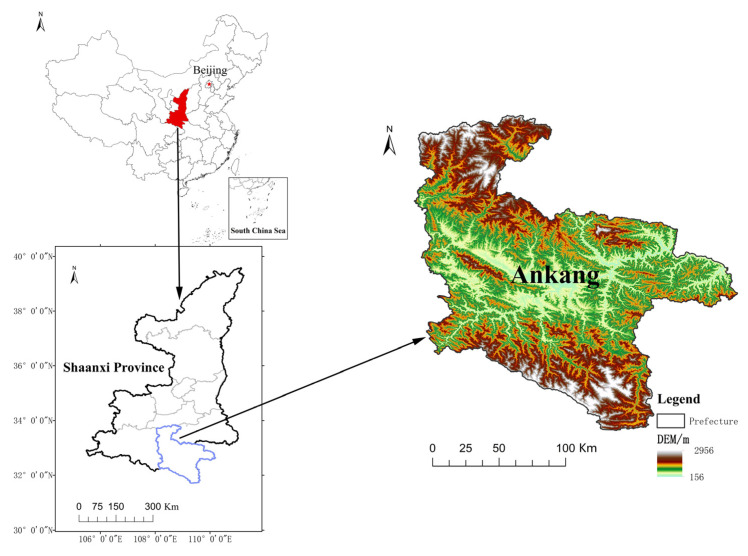
Location of the study area.

**Figure 2 ijerph-20-00694-f002:**
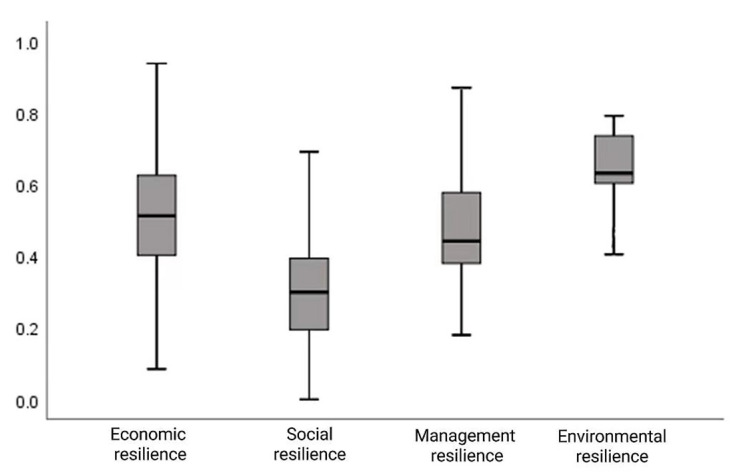
Dimensional boxplot of community resilience.

**Figure 3 ijerph-20-00694-f003:**
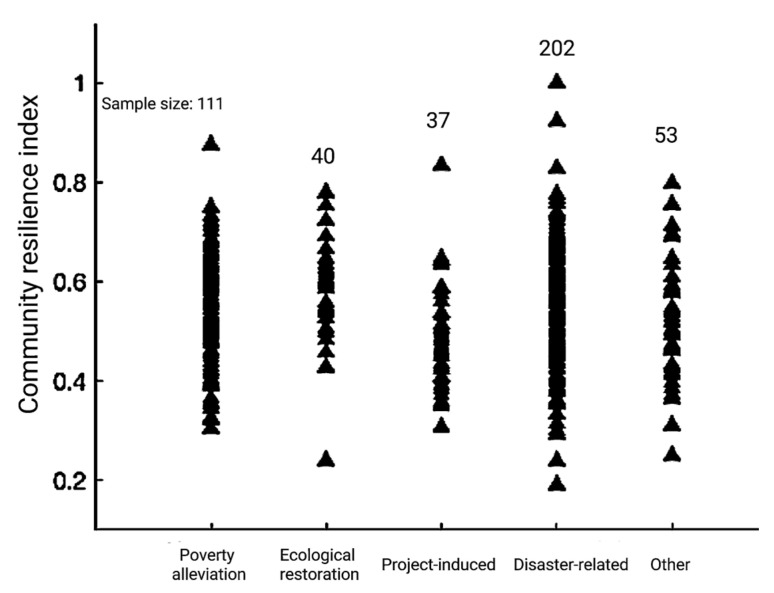
Community resilience index of relocated households by different relocation types.

**Figure 4 ijerph-20-00694-f004:**
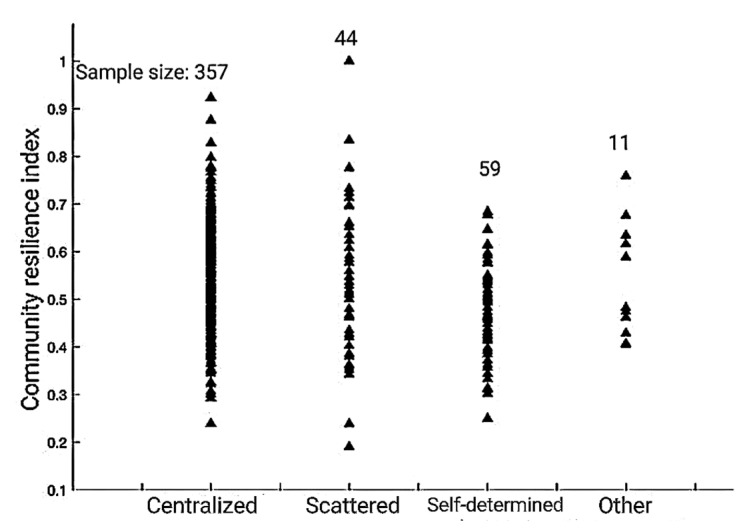
Community resilience index of households with different resettlement modes.

**Figure 5 ijerph-20-00694-f005:**
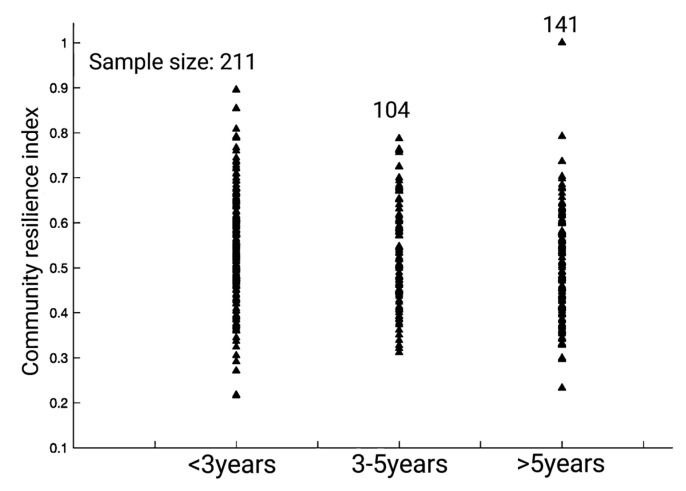
Community resilience index of relocated households at different relocation times.

**Figure 6 ijerph-20-00694-f006:**
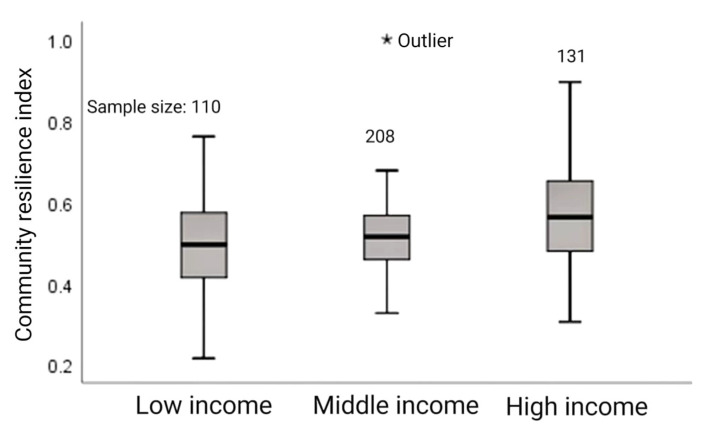
Community resilience index of relocated households at different income levels.

**Figure 7 ijerph-20-00694-f007:**
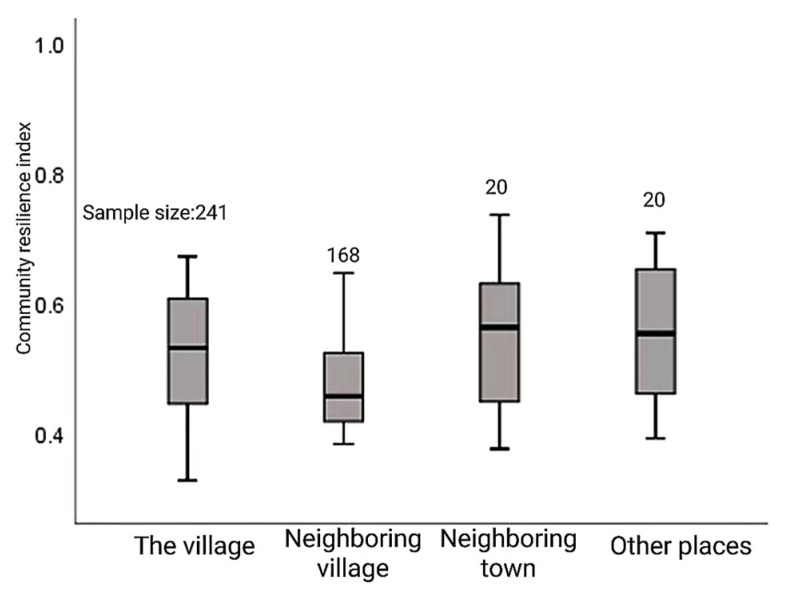
Community resilience index of relocated households in different relocated areas.

**Figure 8 ijerph-20-00694-f008:**
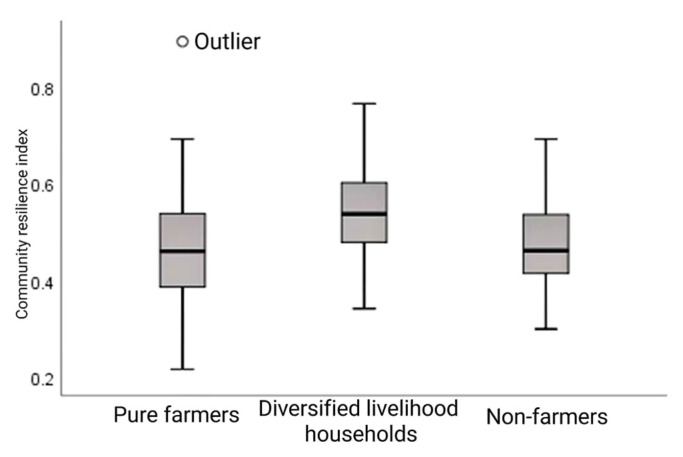
Community resilience index of relocated households in different livelihood types.

**Table 1 ijerph-20-00694-t001:** An index system for measuring the community resilience of rural households relocated in the PAR case.

Objective Layer	Rule Layer	Index Layer	Index Weight	Indicator Meaning and Assignment	Mean Value
A Community resilience	B1 Economic resilience	C_1_ Relocation Housing subsidy	0.0738	Relocation of housing subsidies can effectively alleviate the economic losses brought to the community by the relocation of poverty alleviation and speed up the economic recovery (yuan)	16825.11
		C_2_ Per capita annual net income	0.0542	Quantify the level of community economic growth (yuan)	6025.58
		C_3_ income diversity	0.0519	Reflect the diversity of community economic activities	0.30
		C_4_ Labor force ratio	0.0165	Labor force ratio= Quantity of labor force/family size Number of workers (adults) = family size − number of old people − number of children	0.72
	B2 Social resilience	C_5_ Social help	0.0584	The number of households to whom they can turn in times of difficulty (rumah tangga)	3.82
		C_6_ Social relationships	0.0444	Number of relatives and friends (number)	24.12
		C_7_ Social learning	0.0250	Average number of years of education in the household (number)	26.38
	B3 Management resilience	C_8_ Policy support	0.0341	The number of policies issued by the government to support the improvement of households’ economic conditions (number)	3.20
		C_9_ Social security level	0.0631	The degree of implementation of policies on rights and interests protection in the place of residence (0, 1)	0.28
		C_10_ Government assistance level	0.4356	The reward or subsidy of the government to non-agricultural business activities, and the cash subsidy and in-kind subsidy of the government to the follow-up industry (such as greenhouse, breeding farm construction) (yuan)	190.41
	B4 Environmental resilience	C_11_ Accessibility of public facility	0.0163	Supply of public facilities around the new residence (number)	12.94
		C_12_ Ecological subsidy	0.1267	Government subsidies for converting farmland to forests and ecological public welfare forests (yuan)	510.15

**Table 2 ijerph-20-00694-t002:** Classification of community resilience of relocated households.

Classification Standard	Number of Households	Community Resilience Index Range	Percentage of the Total Sample
Low level	133	0.216–0.432	29.0%
Medium level	196	0.432–0.614	42.7%
High level	130	0.614–1.000	28.3%

**Table 3 ijerph-20-00694-t003:** Descriptive statistics of the sample.

Statistical Variables	Centralized Resettlement		Scattered Resettlement		Self-Determined Resettlement		Other Type Resettlement		
	Mean	Standard Deviation	Mean	Standard Deviation	Mean	Standard Deviation	Mean	Standard Deviation	F
Family size	4.506	1.559	4.093	1.601	4.615	1.601	6.000	2.211	4.060 ***
Average years of education	5.716	2.592	6.790	2.689	5.673	2.271	6.658	1.875	2.861 **
Per capita annual net income	5208.044	6804.382	7081.855	7126.433	9291.360	14,999.028	9619.685	14,099.923	3.532 **
Relocation housing subsidies	19,649.709	18,690.295	8345.000	15,557.140	5696.154	13,022.318	12,440.000	18,002.173	12.820 ***
Income diversity	0.275	0.267	0.443	0.334	0.354	0.310	0.365	0.259	5.544 ***
Labor force ratio	0.709	0.224	0.761	0.227	0.757	0.192	0.739	0.172	1.315
Number of households available for assistance	4.016	5.233	3.929	4.143	2.423	2.607	3.500	2.321	1.641
Sample size	354		43		52		10		

Note: **, *** indicate statistical significance at the level of 5%, and 1%, respectively.

**Table 4 ijerph-20-00694-t004:** The effect of relocation time on the community resilience of relocated Households in the case of PAR.

Variables	Low-Level Community Resilience			High-Level Community Resilience		
	Coef.	Std.	Wald	Coef.	Std.	Wald
Relocation time						
Medium-term	0.753 ***	0.325	5.382	−0.158 *	0.306	0.270
Long-term	1.151 **	0.286	16.160	−0.400 *	0.302	1.769
household size	−0.187 ***	0.083	5.108	0.069 *	0.079	0.774
Income diversification	−1.360 ***	0.508	7.182	1.200 **	0.418	8.237
Net income per capita	−0.152 **	0.099	2.341	0.269 ***	0.092	8.585
Age of household head	−0.030	0.092	0.109	0.298 ***	0.116	6.554
Knowledge of relocation policies	−0.004	0.010	0.130	0.017 *	0.011	2.465
Possibility of loan	−0.003 ***	0.000	10.498	0.000	0.000	0.194
Monthly communication cost of household members	−0.105 ***	0.119	0.774	0.078	0.106	0.533
Constant	1.755 ***	1.110	2.496	−5.322 ***	1.310	16.484
Log-likelihood	−426.315					
Pseudo R^2^	0.123					
Number of observations	459					

Note: “relocation time” takes short-term relocation households as the reference group; *, **, *** indicate statistical significance at the level of 10%, 5%, and 1%, respectively.

**Table 5 ijerph-20-00694-t005:** The effect of relocation type on the community resilience of relocated households in the case of PAR.

Variables	Low-Level Community Resilience			High-Level Community Resilience		
	Coef.	Std.	Wald	Coef.	Std.	Wald
Relocation type						
Poverty alleviation	1.366 **	0.686	3.960	−0.122 **	0.438	0.000
Project-induced	1.840 ***	0.744	6.101	−1.271 ***	0.705	3.240
Disaster-related	1.504 ***	0.663	5.153	−0.004 **	0.410	0.090
Other reasons	2.482 ***	0.705	12.390	−0.489 ***	0.511	0.922
household size	−0.175 ***	0.081	4.666	0.078 ***	0.080	0.960
Income diversification	−1.591 ***	0.514	9.548	1.480 ***	0.443	11.156
Net income per capita	−0.159 **	0.103	2.372	0.266 ***	0.094	8.066
Age of household head	−0.075 **	0.093	0.656	0.286 **	0.117	5.905
Knowledge of relocation policies	−0.003	0.010	0.084	0.015 **	0.011	1.850
Possibility of loan	−0.003	0.001	7.784	0.000 ***	0.000	0.185
Monthly communication cost of household members	−0.131 **	0.117	1.254	0.067 **	0.108	0.384
Constant	0.786 ***	1.253	0.397	−5.250 ***	1.353	15.054
Log-likelihood	−422.274					
Pseudo R^2^	0.132					
Number of observations	459					

Note: “relocation type” takes households relocated by ecological restoration as the reference group; **, *** indicate statistical significance at the level of 5%, and 1%, respectively.

**Table 6 ijerph-20-00694-t006:** The effect of the resettlement mode on the community resilience of relocated households in the case of PAR.

Variables	Low-Level Community Resilience			High-Level Community Resilience		
	Coef.	Std.	Wald	Coef.	Std.	Wald
Resettlement mode						
Scattered resettlement	1.680 ***	0.458	13.469	−0.276 ***	0.475	0.336
Self-determined resettlement	1.984 ***	0.393	25.402	−0.571 ***	0.479	1.416
Other type	1.245	0.904	1.904	0.192	0.818	0.053
household size	−0.168 ***	0.085	3.920	0.076 ***	0.081	0.846
Income diversification	−2.009 ***	0.544	13.616	1.214 ***	0.431	7.952
Net income per capita	−0.216 ***	0.105	4.244	0.257 ***	0.091	7.952
Age of household head	−0.117 **	0.096	1.464	0.323 ***	0.118	7.453
Knowledge of relocation policies	−0.000	0.011	0.000	0.016 **	0.011	2.341
Possibility of loan	−0.003 **	0.001	11.156	0.000 ***	0.000	0.325
Monthly communication cost of household members	−0.017 ***	0.120	0.792	0.083 ***	0.107	0.593
Constant	2.667 ***	1.152	5.336	−5.622 ***	1.340	17.640
Log-likelihood	−414.766					
Pseudo R^2^	0.147					
Number of observations	459					

Note: “resettlement mode” takes centralized resettlement households as the reference group; **, *** indicate statistical significance at the level of 5%, and 1%, respectively.

## Data Availability

Data is available on request due to privacy/ethical restrictions.

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
