# Peer review of "Measuring Community Resilience and Its Determinants: Relocated Vulnerable Community in Western China"

_ijerph, 2022, doi:10.3390/ijerph20010694_

Round 1
Reviewer 1 Report
Discussion on community resilience is a very interesting study. The authors quantitatively discuss the drivers of resilience based on survey data from western China. The writing of the paper is fluent and can attract a wide range of readers. I suggest it be published after minor revision. Specific revision suggestions are as follows:
(1) In the method part, I suggest that the authors supplement the calculation process of entropy weight method.
(2) Pictures. I suggest that the authors add illustrations to improve the readability of the pictures.
(3) In the discussion part, I suggest that the author mainly explain the differences between the findings of this study and the existing studies. And put the policy recommendations in the conclusion part.
Author Response
Response to Reviewer 1 Comments
Point 1: In the method part, I suggest that the authors supplement the calculation process of entropy weight method.
Response 1: Thank you very much for this constructive recommendation. We added the calculation process of entropy method at the end of Section 2.3.
Point 2: Pictures. I suggest that the authors add illustrations to improve the readability of the pictures.
Response 2: Thank you very much for the useful comment. We have added a map of the study area, and added illustrations of sample size to the Figure 3—8 .
Point 3: In the discussion part, I suggest that the author mainly explain the differences between the findings of this study and the existing studies. And put the policy recommendations in the conclusion part.
Response 3: Thank you very much for the comments. We moved the section on recommendations from the Discussion section to the Conclusion section.

Reviewer 2 Report
The findings show that all forms of relocation (poverty alleviation, project-induced, disaster-related, and other reasons) have a negative impact on both both low and high resilience. These findings should be expanded on in the discussion section. Further more the practical implication on these matter should clearly describe in the conclusion section. In addition to the conclusion section, suggestions for future research are necessary to strengthen this manuscript.
1. What is the main question addressed by the research? - The primary research questions in this study are whether relocation strategies and other related factors have an impact on vulnerable communities with low and high resilience.
2. Do you consider the topic original or relevant in the field? Does it
address a specific gap in the field? The topic is considered relevant in the context of China's relocation strategy for vulnerable communities. However, the title should be changed to "Measuring Community Resilience and its Determinants: Relocated Vulnerable Community in Western China."
3. What does it add to the subject area compared with other published
material? In terms of measuring community resilience, there is no additional info. However, this paper has good empirical evidence for the policy framework.
4. What specific improvements should the authors consider regarding the
methodology? What further controls should be considered? a.Insert a map of the study location based on the types of resettlement. b. When the survey was conducted, how the sample size was calculated, and how the sample was chosen c. Insert community resilience index formula. d. Insert the community resilience classification range based on the community resilience index. 5. Are the conclusions consistent with the evidence and arguments presented
and do they address the main question posed? Overall, the conclusions are reasonable in the context of the evidence and arguments presented.
6. Are the references appropriate? Reasonable enough.
7. Please include any additional comments on the tables and figures. a) Insert the relocation duration and describe the PAR's pre, during, and post relocation planning at page 2, line 52. b) Please refer my comments in the reviewer's comments
Author Response
Point 1: What is the main question addressed by the research? The primary research questions in this study are whether relocation strategies and other related factors have an impact on vulnerable communities with low and high resilience.
Point 2: Do you consider the topic original or relevant in the field? Does it address a specific gap in the field? The topic is considered relevant in the context of China's relocation strategy for vulnerable communities. However, the title should be changed to "Measuring Community Resilience and its Determinants: Relocated Vulnerable Community in Western China."
Response 2: Thank you very much for this constructive recommendation. The title of this paper has been changed to “Measuring Community Resilience and its Determinants: Relocated Vulnerable Community in Western China”.
Point 3: What does it add to the subject area compared with other published material? In terms of measuring community resilience, there is no additional info. However, this paper has good empirical evidence for the policy framework.
Point 4: What specific improvements should the authors consider regarding the methodology? What further controls should be considered? a.Insert a map of the study location based on the types of resettlement. b. When the survey was conducted, how the sample size was calculated, and how the sample was chosen c. Insert community resilience index formula. d. Insert the community resilience classification range based on the community resilience index.
Response 4: Thank you very much for this constructive recommendation.
a.We added a map of the study area locations in Section 2.1.
- We added the research time, sample size calculation method, and sample selection method in Section 2.1.
- The community resilience index is calculated by the entropy method.We added the calculation process of entropy method at the end of Section 2.3.
- We have added table 2 on the range of community resilience indices in section 2.4.
Point 5: Are the conclusions consistent with the evidence and arguments presented and do they address the main question posed? Overall, the conclusions are reasonable in the context of the evidence and arguments presented.
Point 6: 6. Are the references appropriate? Reasonable enough.
Point 7: Please include any additional comments on the tables and figures. a) Insert the relocation duration and describe the PAR's pre, during, and post relocation planning at page 2, line 52. b) Please refer my comments in the reviewer's comments
Response 7: Thank you very much for this constructive recommendation. We insert the migration duration on page 2, line 52, and describe the pre-migration, migration, and post-migration planning for PAR.

Reviewer 3 Report
Overall, it’s an interesting study. This manuscript attempted to propose a method for evaluating the community resilience by identified determinants from four aspects of economic resilience, social resilience, management resilience, environment resilience. The authors have done deep investigation in villages and the problems the authors mentioned on PAR did exist. Yet, while, as the authors said, “there are no unified regulation for selecting resilience indicators”, there is a lack of indicators for the economic burden, energy supply and the frequency of various disasters. The resilience is to describe a kind of ability to overcome some bad influences. In your proposed method, this kind of ability was not presented. In details, I cannot judge whether the investigated households can afford by their income or government subsidies for their daily life or stay healthy in a cold winter without gas supply. Since the information such as living costs, energy infrastructure condition, building structure, even the pictures of the villages were not presented, I have to reject this manuscript.
Here are some tips for the authors:
1. What is the objective of community resistance, in other words, please clarify what do you want the community to resistant: the possible disaster? Back to poverty? The issue was addressed to is because as the objective changes, the determinants of community resilience may also change.
2. There were some studies on normal community resistance presented, but it is unclear how the rural community may be different. Please explain the specialty in rural area, especially in rural PAR area.
3. The content from lines 115-126, should not be in Section 2, consider to move it to Section 1.
4. Please explain more details of the study area selection reason: the author chose 10 villages in Ziyang County, Hanbin District, and Ningshan County of Ankang Prefecture, as the samples. You mention the reason as they were “with concentrated ecological policies and prominent nature conservation problem”. Please show us the proof with data, for example 30% higher / lower than average level.
5. Study area information: please show us the description of the 10 villages in terms of t relocation types, modes, time, income level, relocated area, etc. It is better to show them in a table. Moreover, you should define some of the dimensions, such as the income level was classified to 3 levels, how to define the high, medium, low?
6. Table 1, column of “Mean value “, what are the counting Units respectively? Income by RMB? Yearly? Monthly? If the values were obtained from the survey, please show us the selection range, from 0-1? From 1-10?
7. Figures 2-7, sample size should be addressed in every single dimension.
Author Response
Response to Reviewer 3 Comments
Point 1: What is the objective of community resistance, in other words, please clarify what do you want the community to resistant: the possible disaster? Back to poverty? The issue was addressed to is because as the objective changes, the determinants of community resilience may also change.
Response 1: Thank you very much for the comments. Community resilience is studied to resist the decline in community living standards after community farmers change their place of residence triggered by environmental changes, major construction projects, and natural disasters.
Point 2: There were some studies on normal community resistance presented, but it is unclear how the rural community may be different. Please explain the specialty in rural area, especially in rural PAR area.
Response 2: Thank you very much for the comments. Most of the areas involved in the poverty alleviation resettlement project for are ecologically fragile areas, seismically active zones, geological disaster-prone areas, and areas where the promotion of production or employment through local assistance is still unable to lift the farmers out of poverty. The focus of the project is to help poor households with relatively deep poverty.
Point 3: The content from lines 115-126, should not be in Section 2, consider to move it to Section 1.
Response 3: Thank you for your suggestion. We have moved lines 115-126 to section 1.
Point 4: Please explain more details of the study area selection reason: the author chose 10 villages in Ziyang County, Hanbin District, and Ningshan County of Ankang Prefecture, as the samples. You mention the reason as they were “with concentrated ecological policies and prominent nature conservation problem”. Please show us the proof with data, for example 30% higher / lower than average level.
Response 4: The cause of relocation in Ankang was the "7-18" mudslide disaster in 2010. From the initial disaster relocation to the current poverty alleviation relocation. The relocated population accounts for one-third of the population and has a high rate of centralized relocation. Therefore, it is suitable as a sample for relocation research.
Point 5: Study area information: please show us the description of the 10 villages in terms of t relocation types, modes, time, income level, relocated area, etc. It is better to show them in a table. Moreover, you should define some of the dimensions, such as the income level was classified to 3 levels, how to define the high, medium, low?
Response 5: Thank you very much for the comments. We did not summarize the specific characteristics of each village because our research was conducted on specific farm households and the study area was Ankang City, where the sample did not vary much between each village.
Point 6: Table 1, column of “Mean value “, what are the counting Units respectively? Income by RMB? Yearly? Monthly? If the values were obtained from the survey, please show us the selection range, from 0-1? From 1-10?
Response 6: Thank you very much for this constructive recommendation. We have supplemented the units of measure for the variables in Table 1.
Point 7: Figures 2-7, sample size should be addressed in every single dimension.
Response 7: Thank you very much for this constructive recommendation. We have labeled the sample size of each dimension in the figure.

Reviewer 4 Report
Dear authors, much work has been done to the manuscript, however, here are some of my concerns.
1) CR measurement is no easy work, to me, the index calculation of CR from four dimensions in the paper is more like a comprehensive evaluation of communities, grading their scores. Resilience needs more interpretation, especially a comparison before and after relocation, from perspective of economic, social, environmental, and so on. Just like ref No.35 said, CR is a dynamic process, how can you measure it with cross-section data.
2) I'm also worried about certain index. In line 181-184, authors provide the idea that social relationships "can be calculated by behaviors such as sending New Year's greeting...and chatting...", I believe there must be some other ways to quantify social relationship more properly.
3) Should multicollinearity and endogeneity be concerned since regression model is employed? Moreover, some characteristic variables in Table 3-5 to explain the resilience are the same as indexes to measure CR in Table 1, such as the education index, I also find improper.
Author Response
Response to Reviewer 4 Comments
Point 1: CR measurement is no easy work, to me, the index calculation of CR from four dimensions in the paper is more like a comprehensive evaluation of communities, grading their scores. Resilience needs more interpretation, especially a comparison before and after relocation, from perspective of economic, social, environmental, and so on. Just like ref No.35 said, CR is a dynamic process, how can you measure it with cross-section data.
Response 1: Thank you very much for this constructive recommendation. We are also aware of this problem and have raised this shortcoming in our discussion, but due to the epidemic there is no way to conduct large-scale follow-up research activities, so we have to follow the study of cross-sectional data. However, the group has previously published several papers on the factors influencing livelihood resilience, and the cross-sectional data have been recognized by several international academic journals, so we have continued to use the cross-sectional data this time. In the future academic research we you will also keep in mind your valuable comments and continue to improve the scientific data.
Liu, W.; Li, J.; Xu, J. Effects of Disaster-Related Resettlement on the Livelihood Resilience of Rural Households in China. Int. J. Disaster Risk Reduct. 2020, 49, 10.[http://doi.org/10.1016/j.ijdrr.2020.101649]
Liu, W.; Li, J.; Ren, L.; Xu, J.; Li, C.; Li, S. Exploring Livelihood Resilience and Its Impact on Livelihood Strategy in Rural China. Social Indicators Research 2020, 150, (3), 977-998.
Point 2: I'm also worried about certain index. In line 181-184, authors provide the idea that social relationships "can be calculated by behaviors such as sending New Year's greeting...and chatting...", I believe there must be some other ways to quantify social relationship more properly.
Response 2: Thank you very much for the useful comment. Social relationships refer to the number of relatives and friends who remain in close contact with the household, which can be calculated by the act of exchanging gifts and sending and receiving text messages with friends and relatives.
Point 3: Should multicollinearity and endogeneity be concerned since regression model is employed? Moreover, some characteristic variables in Table 3-5 to explain the resilience are the same as indexes to measure CR in Table 1, such as the education index, I also find improper.
Response 3: Thank you very much for this constructive recommendation. To avoid the problem of multicollinearity, we replace these four variables Average years of education、Social relations
Number of households available for assistance andAccessibility to public facilities with Net income per capita、Age of household head、Knowledge of relocation policies and Monthly communication cost of household members。
